# Eupatin, a Flavonoid, Inhibits Coronavirus 3CL Protease and Replication

**DOI:** 10.3390/ijms24119211

**Published:** 2023-05-24

**Authors:** Yea-In Park, Jang Hoon Kim, Siyun Lee, Ik Soo Lee, Junsoo Park

**Affiliations:** 1Division of Biological Science and Technology, Yonsei University, Wonju 26493, Republic of Korea; 2Department of Herbal Crop Research, National Institute of Horticultural & Herbal Science, RDA, Eumsung 27709, Republic of Korea; 3Km Covergence Research Division, Korea Institute of Oriental Medicine, Daejeon 34054, Republic of Korea

**Keywords:** SARS-CoV-2, flavonoid, 3CL protease, *Inula japonica*

## Abstract

The coronavirus disease 2019 (COVID-19) pandemic has caused more than six million deaths worldwide since 2019. Although vaccines are available, novel variants of coronavirus are expected to appear continuously, and there is a need for a more effective remedy for coronavirus disease. In this report, we isolated eupatin from *Inula japonica* flowers and showed that it inhibits the coronavirus 3 chymotrypsin-like (3CL) protease as well as viral replication. We showed that eupatin treatment inhibits SARS-CoV-2 3CL-protease, and computational modeling demonstrated that it interacts with key residues of 3CL-protease. Further, the treatment decreased the number of plaques formed by human coronavirus OC43 (HCoV-OC43) infection and decreased viral protein and RNA levels in the media. These results indicate that eupatin inhibits coronavirus replication.

## 1. Introduction

Severe acute respiratory syndrome coronavirus 2 (SARS-CoV-2) caused the coronavirus disease 2019 (COVID-19) pandemic, resulting in more than 6 million deaths worldwide [1]. Recently, vaccines for COVID-19 have been developed; however, novel variants of coronavirus have continuously appeared, and various therapies for coronavirus diseases are required [2]. The coronavirus contains several major drug targets, including 3 chymotrypsin-like protease (3CL-protease), papain-like protease (PL-protease), and RNA-dependent RNA polymerase [3]. Among them, 3 chymotrypsin-like protease is a popular target for drug development, and recently, Paxlovid was developed to treat COVID-19 [4,5].

Natural products have been used medicinally worldwide since ancient times. They contain several secondary metabolites with the potential to prevent or treat various human diseases [6,7]. Plant-derived secondary metabolites include alkaloids, polyphenols, flavonoids, terpenoids, and phytosterols [8]. Flavonoids are phenyl-substituted chromones with a C_6_-C_3_-C_6_ carbon skeleton consisting of two benzene rings (rings A and B) linked with a three-carbon heterocyclic ring (ring C) [9]. They are widespread in the plant kingdom, and more than 9000 flavonoids have been identified as secondary metabolites in plants [10]. Flavonoids are of great interest in medicine due to their diverse pharmacological properties, including antioxidant, anti-inflammatory, analgesic, anticancer, antibacterial, antifungal, and antiviral activities [11]. 

While searching for effective SARS-CoV-2 3CL-protease and coronavirus inhibitors from natural sources, we isolated and identified eupatin as the main active compound in the flower extract of *Inula japonica* Linnaeus (Asteraceae) [12]. A recent report demonstrated that eupatin exerts anti-inflammatory effects by suppressing the expression of phospho-p65, inducible nitric oxide synthase, and cyclooxygenase-2 [13]. In this study, we demonstrated that eupatin inhibits the activity of SARS-CoV-2 3CL-protease, and computational models support the hypothesis that it interacts with 3CL-protease. We used the human coronavirus OC43 (HCoV-OC43) model coronavirus system [14] and demonstrated that eupatin treatment inhibits coronavirus replication of coronavirus. 

## 2. Results

### 2.1. Isolation and Identification of Eupatin

Chromatographic separation of the ethanol extract of *I. japonica* flowers led to the isolation of an amorphous yellow powder with the molecular formula C_18_H_16_O_8_, as established with ESI-MS, based on a molecular ion peak at *m*/*z* 359.07 [M − H]^−^. The characteristic UV absorption maxima (255 and 345 nm) of this compound suggested the presence of a flavonol skeleton (Appendix A). Its ^1^H-NMR spectrum showed an aromatic singlet at δ_H_ 6.85 (1H, s, H-8) and ABX aromatic system protons at δ_H_ 7.58 (1H, dd, *J* = 9.2, 2.0 Hz, H-6′), 7.56 (1H, d, *J* = 2.0 Hz, H-2′), and 7.10 (1H, d, *J* = 9.2 Hz, H-5′) (Appendix A). The signals at δ_H_ 3.90, 3.86, and 3.80 (each 3H, s) were assigned to the methoxy groups at C-6, C-7, and C-4′, respectively. The ^13^C-NMR spectrum showed that this compound contained 18 carbons, of which 15 carbon signals were assigned to the 3-*O*-substituted flavonol structure, and three carbon signals (δ_C_ 59.7, 56.3, and 55.7) were assigned to three methoxy groups substituted in the flavonol structure (Appendix A). Finally, this compound was identified as eupatin by comparing its spectral data with those reported in the literature [6] (Figure 1A).

### 2.2. The Inhibitory Activity of Eupatin on SARS-CoV-2 3CL-Protease

The enzyme inhibitory activity was evaluated in vitro prior to the coronavirus inhibition experiment. Eupatin inhibited the catalytic reaction of the 3CL-protease with the substrate in a concentration-dependent manner at 12.5–100 μM concentration. Based on this, the calculated IC_50_ value of the compound was 34.9 ± 1.7 μM (Figure 1B). The binding between the compound and enzyme was analyzed using enzyme kinetics. As indicated in Figure 1C and Table 1, this inhibitor was confirmed to be binding to the active site of 3CL-protease as a competitive inhibitor, with *k*_i_ values of 30.0 μM (Figure 1D).

### 2.3. The Prediction of Binding Pose between Eupatin and 3CL-Protease

This study visualized the binding of eupatin in the active site of the 3CL-protease based on computational chemistry (Figure 1E). The autodock program tracked their optimal bonding energy, and a most stable value of −7.47 kcal/mol autodock score was calculated (Table 1). The 5-hydroxyl group of the inhibitor maintained a hydrogen bond at a 2.91 Å distance with Cys145 (Figure 1F, Table 1). Moreover, the 3′-hydroxyl group formed four hydrogen bonds at 2.98, 2.68, 2.79, and 2.95 Å distance with Arg188, Thr190, and Gln192, respectively.

### 2.4. Molecular Dynamics Study of Eupatin with SARS-CoV-2 3CL-Protease

To analyze the detailed interaction between the ligand and receptor, molecular dynamics studies were performed on the complex with molecular docking. They showed a stable and fluid motion while maintaining −2.3 × 10^6^ kJ/mol potential energy for 30 ns (Figure 2A,B). The protein-based root-mean-square derivation (RMSD) was below 0.5 nm, according to the protein movement over time compared to the initial protein (Figure 2C). In root-mean square-fluctuations (RMSF), some amino residues also had motions of 0.3–0.5 nm (Figure 2D). They mainly formed 2–4 hydrogen bonds, occasionally 0 and 5–7 (Figure 2E). Analyzing these simulation times at 3 ns intervals confirmed that four amino residues (His41, Glu166, Thr190, and Gln192) form major hydrogen bonds. His41 maintained a distance of ~4 Å to the 4-keton of eupatin for 30 ns. Glu166 and Gln192 maintained a distance of ~3 Å from the 1-ether and 3′-hydroxyl groups, respectively. Notably, Thr190 formed two hydrogen bonds with the 3′-hydroxyl group and maintained ~2.5 a distance of 0 to 28 ns.

### 2.5. Eupatin-Treatment-Reduced Coronavirus Replication

Since eupatin inhibits 3CL protease activity, we hypothesized that eupatin reduces human coronavirus replication. A plaque formation assay was used to examine the effect of eupatin on coronavirus replication. RD cells were treated with mock or eupatin and infected with the indicated dilutions of HCoV-OC43-infected cell media. Eupatin treatment decreased plaque formation in RD cells in a dose-dependent manner (Figure 3A,B). In addition, we examined the cytotoxicity of eupatin in RD cells. RD cells were treated with the indicated concentrations of eupatin for 24 h, and cell viability was measured. While a high-dose eupatin (>10 μM) treatment showed cytotoxicity, a lower dose did not significantly affect the viability of RD cells (Figure 3C).

### 2.6. Eupatin Inhibits Coronavirus Protein Expression and Replication

As eupatin treatment reduced plaque formation, we examined the expression levels of coronavirus proteins. RD cells were treated with mock or eupatin and infected with HCoV-OC43. Eupatin treatment did not change the viral protein expression in the infected cells; however, it dramatically reduced the level of HCoV-OC43 protein in the conditioned media (Figure 4A,B). Since the coronavirus protein in the conditioned media was derived from coronavirus particles, these results indicated that eupatin treatment decreases the level of coronavirus particles in the media. 

Then, we examined the expression of HCoV-OC43 RNA in infected cells and media using quantitative RT-PCR. RD cells were treated with mock or eupatin and infected with HCoV-OC43. We evaluated the level of coronavirus RNA in the conditioned media 72 h after infection. The HCoV-OC43 M, N, and RNA-dependent RNA polymerase (RDRP) genes were used to evaluate the level of coronavirus RNA. Similar to protein expression, eupatin treatment did not affect the level of coronavirus RNA in infected cells. However, eupatin treatment effectively decreased the level of coronavirus RNA (M, N, and RdRp genes) in the conditioned media (Figure 4C). These results indicated that eupatin treatment decreases the level of coronavirus in the media.

## 3. Discussion

Humans have used natural products for millennia to treat diseases and health disorders [15]. Some medicinal plants are useful antiviral agents [16]. Secondary metabolites derived from these medicinal plants can play an essential role in antiviral drug discovery by providing lead scaffolds that can be optimized by synthetic and medicinal chemists. *I. britannica* is one of the most commonly used plant species in traditional Chinese medicine for treating digestive disorders, bronchitis, and various types of inflammation [17,18]. The flowers of this plant contain various secondary metabolites such as steroids, terpenoids (sesquiterpenes, diterpenes, and triterpenoids), phenolics, and flavonoids [17]. Flavonoids are frequently reported as major active ingredients that exhibit enzyme inhibitory, antioxidant, and cytotoxic activities [17]. Structurally, eupatin contains a 3-*O*-substituted flavonol skeleton with three methoxy groups. It has been reported to possess anti-inflammatory and acetylcholinesterase-inhibitory effects [13,19].

In a recent report, we confirmed the inhibitory effects on SARS-CoV-2 3CL-protease and HCoV-OC43 of hispidulin, patuletin, and nepetin from *I. britannica* flowers [20]. Moreover, recent computer simulation studies reported that eupatin could bind to 3CL-protease and ACE2 [21]. Eupatin, a flavonoid isolated from the flower extract of *I. japonica*, exhibited considerable 3CL-protease inhibitory activity. Eupatin stably fits into the active site as a competitive inhibitor. The molecular simulation revealed the key amino acids for this inhibition as His41, Glu166, Thr190, and Gln192. We showed that eupatin treatment resulted in decreased coronavirus replication. Notably, eupatin treatment did not significantly change the levels of coronavirus protein and RNA in infected cells; however, eupatin treatment decreased the levels of both protein and RNA in the media. As the viral protein and RNA in the media are from viral particles, these results suggest that eupatin inhibits the release of coronavirus from the cells. The total amounts of viral protein and RNA (media and cells) were reduced by eupatin treatment. Oseltamivir (Tamiflu) also inhibits the spread of the influenza virus by inhibiting neuraminidase, and further studies are required to elucidate the detailed mechanism of eupatin [22].

High concentrations of single compounds isolated from plants often result in cytotoxicity. Similarly, we found that a high concentration of eupatin (≥10 μM) resulted in significant cytotoxicity; however, lower concentrations (0.5–5 μM) inhibit coronavirus replication significantly. Therefore, eupatin concentration should be considered in further experiments to prevent cytotoxicity. Alternatively, the development of derivative compounds with low cytotoxicity is desirable to improve their efficacy.

In this report, we did not use SARS-CoV-2 due to the stringency of regulations. However, HCoV-OC43 and SARS-CoV-2 belong to the beta coronavirus family and share many biological characteristics. Therefore, the knowledge obtained from eupatin and HCoV-OC43 will be helpful in the development of novel drugs for coronaviruses.

## 4. Materials and Methods

### 4.1. General Experimental Procedures

Ultraviolet (UV) spectra were recorded using a JASCO V-550 UV/VIS spectrometer (Jasco, Japan). ^1^H (400 MHz) and ^13^C (100 MHz) nuclear magnetic resonance (NMR) spectra were obtained using a Bruker DRX-400 spectrometer (Bruker, Ettlingen, Germany) with tetramethylsilane as the internal standard. Electrospray ionization-mass spectrometry (ESI-MS) was performed on a Shimadzu LCMS-IT-TOF spectrometer (Shimadzu Co., Kyoto, Japan). Thin-layer chromatography (TLC) analysis was performed on silica gel 60 F254 and RP-18 F254S plates (0.25 mm layer thickness; Merck, Darmstadt, Germany). Column chromatography was performed using silica gel (60 A, 70–230, or 230–400 mesh ASTM; Merck, Darmstadt, Germany) and reversed-phase silica gel (ODS-A 12 nm S-150, S-75 μm; YMC Co., Kyoto, Japan). Spots were detected with UV light (254 nm) and sprayed with 10% H_2_SO_4_, followed by heating. 3CLpro was purchased from Sigma Aldrich (St. Louis, MO, USA). 3Clpro Substrate was synthesized with Anygen (Gwangju, Republic of Korea).

### 4.2. Plant Material

*I*. *japonica* flowers were purchased from a traditional herbal medicine store in Daejeon, Republic of Korea, in August 2020 and identified by Prof. Y. H. Kim (College of Pharmacy, Chungnam National University, Republic of Korea). A voucher specimen (IJ2020-030) has been deposited at the herbarium of the Korea Institute of Oriental Medicine, Republic of Korea.

### 4.3. Extraction and Isolation

Air-dried *I*. *japonica* flowers (200 g) were extracted with ethanol (2 L) at 80 °C for 3 h, filtered, and concentrated to yield an ethanol extract (10 g, 5% yield). The extract was subjected to silica gel column chromatography (50 × 8.5 cm) using a methylene chloride-methanol (1:0→0:1) gradient solvent system. The column chromatographic fractions were combined to obtain four fractions (A–D) based on the TLC data. Fraction A was subjected to reversed-phase silica gel column chromatography (50 × 5 cm) using a methanol-water (20:80→100:0) gradient solvent system to yield five subfractions (A1–A5). Fraction A2 was further chromatographed on a reversed-phase silica gel column (60 × 4 cm) using a methanol-water (60:40→80:20) gradient to generate six subfractions (A2.1–A2.6). Fraction A2.3 was then chromatographed separately in a reversed-phase silica gel column (60 × 2.5 cm) using a methanol-water (60:40→80:20) gradient to yield eupatin (32 mg).

Eupatin: Yellow amorphous powder. UV (MeOH) λ_max_ nm: 255, 345. ESI-MS *m*/*z*: 359.07 [M − H]^−^. ^1^H-NMR (400 MHz, DMSO-*d*_6_): δ_H_ 7.58 (1H, dd, *J* = 9.2, 2.0 Hz, H-6′), 7.56 (1H, d, *J* = 2.0 Hz, H-2′), 7.10 (1H, d, *J* = 9.2 Hz, H-5′), 6.85 (1H, s, H-8), 3.90, 3.86, 3.80 (each 3H, s, 6, 7, 4′-OCH_3_). ^13^C-NMR (100 MHz, DMSO-*d*_6_): δ_C_ 150.2 (C-2), 137.9 (C-3), 178.2 (C-4), 148.9 (C-5), 129.7 (C-6), 155.4 (C-7), 90.9 (C-8), 154.6 (C-9), 105.6 (C-10), 122.5 (C-1′), 115.1 (C-2′), 145.7 (C-3′), 146.3 (C-4′), 111.9 (C-5′), 120.3 (C-6′), 59.7 (3H, s, 6-OCH_3_), 56.3 (3H, s, 7-OCH_3_), 55.7 (3H, s, 4′-OCH_3_).

### 4.4. 3CL-Protease Inhibition Assay

The assay was performed as described previously [20]. A total of 200 μL of a mixture of the enzyme (13.3 μg/mL), substrate (100 μM), and inhibitor (methanol or eupatin) was added to a 96-well plate. The mixture (enzyme-substrate-inhibitor) was incubated at 37 °C for 30 min. The amount of product was detected using a fluorescence spectrophotometer to measure 530 nm emission with 340 nm excitation. The difference in the amounts of product in the methanol and eupatin mixtures was calculated as the inhibitory activity (%). Molecular docking and dynamics were analyzed as previously described [20].

### 4.5. Cell Culture, Human Coronavirus Infection to Cells, and Cell Viability Assay

RD cells were maintained in Dulbecco’s modified Eagle medium (DMEM; Welgene, Seoul, Republic of Korea) containing 10% fetal bovine serum (FBS; Thermo Fisher Scientific, Waltham, MA, USA) and 1% antibiotic-antimycotic solution (Welgene, Seoul, Republic of Korea). RD cells were infected with HCoV-OC43 at the indicated dilution of virus-infected cell media, and the infected cells were maintained in minimum essential medium (MEM) containing 2% FBS and 1% penicillin-streptomycin (Welgene). The cell viability was assessed using EzCytox cell viability assay kits (DoGen, DAEILLAB, Seoul, Republic of Korea). RD cells were seeded in the wells of a 24-well plate and incubated with the indicated concentrations of eupatin for 24 h. RD cells were obtained from the Korean Cell Line Bank (KCLB, Seoul, Republic of Korea). The plaque formation assay was performed as previously described [20].

### 4.6. Quantitative Reverse Transcription-Polymerase Chain Reaction (RT-PCR)

Quantitative RT-PCR was used to measure the level of HCoV-OC43 RNA in infected cell media [23]. For quantitative RT-PCR, media were harvested, and RNA was extracted using Trizol (Thermo Fisher Scientific) as per the manufacturer’s instructions and subjected to RT-PCR using the StepOnePlus Real-Time PCR system (Thermo Fisher Scientific). HCoV-OC43 RdRp(RNA dependent RNA polymerase) gene was amplified using the forward primer 5′-GAGTGTAGATGCCCGTCTCG-3′ and reverse primer 5′-TGTGGCACACGACTACCTTC-3′. HCoV-OC43 M gene was amplified using the forward primer 5′-ACGGTCACAATAATACGCGGT-3′ and reverse primer 5′-GGGTTGATGGCAGTCGGTAA-3′. HCoV-OC43 N gene was amplified using the forward primer 5′-AGGATGCCACCAAACCTCAG-3′ and reverse primer 5′-TGGGGAACTGTGGGTCACTA-3′. 

### 4.7. Western Blots

Western blotting was used to measure the level of coronavirus protein in cells and media. For western blotting, the cells and media were harvested and resuspended in cell lysis buffer (150 mM NaCl, 50 mM HEPES, and 1% NP40 at pH 7.5) containing a protease inhibitor cocktail (Roche, Basel, Switzerland). Equal amounts of proteins were subjected to western blotting, and virus proteins were detected with a 1:5000 dilution of primary anti-HCoV-OC43 antibody (MAB9012, Sigma-Aldrich, St. Louis, MO, USA). Images were acquired using a Chemi-Doc XRS+ (Bio-Rad, Richmond, CA, USA).

### 4.8. Statistical Analysis

Results are shown as mean ± standard error values. Differences between groups were evaluated with a 2-tailed Student’s *t*-test using Excel software (Microsoft, Redmond, WA, USA). *p* < 0.05 was considered significant.

## Figures and Tables

**Figure 1 ijms-24-09211-f001:**
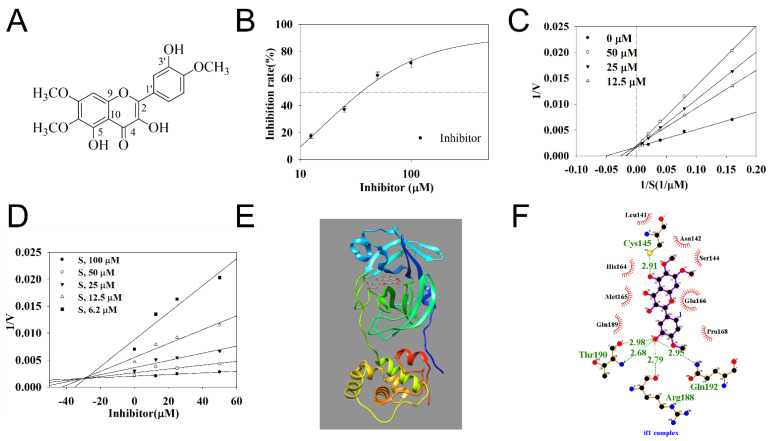
Chemical structure of eupatin (**A**) isolated from *I. britannica* flowers. The inhibitory activity of inhibitor on 3CL-protease (**B**). Lineweaver–Burk and Dixon plots were constructed from the inhibition of 3CL-protease by the inhibitor (**C**,**D**). The best docking position of the inhibitor is indicated with the green dotted line representing hydrogen bonds of inhibitor with key residues of 3CL-protease(**E**,**F**).

**Figure 2 ijms-24-09211-f002:**
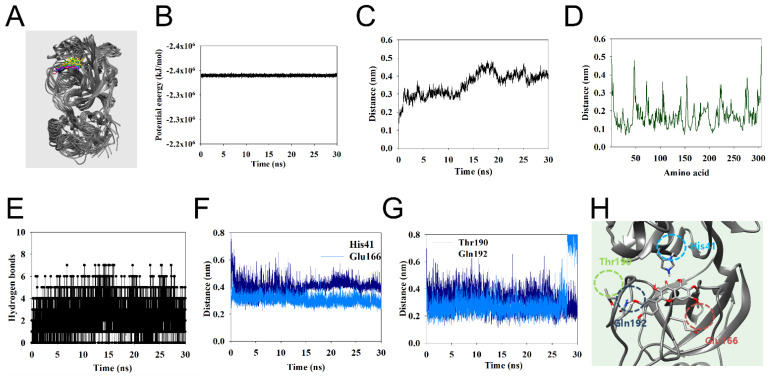
The superpositions of 3CL-protease with eupatin for the simulation time (red: 0 ns, orange: 3 ns, yellow: 6 ns, green: 9 ns, cyan: 12 ns, blue: 15 ns, cornflower blue: 18 ns, purple: 21 ns, hot pink: 24 ns, magenta: 27 ns, and black: 30 ns) (**A**), The potential energy (**B**), RMSD (**C**), RMSF (**D**), hydrogen bond numbers (**E**), and distance between the inhibitor and four residues (**F**,**G**) of the simulation calculated during 30 ns. The complex of key residues with eupatin (**H**).

**Figure 3 ijms-24-09211-f003:**
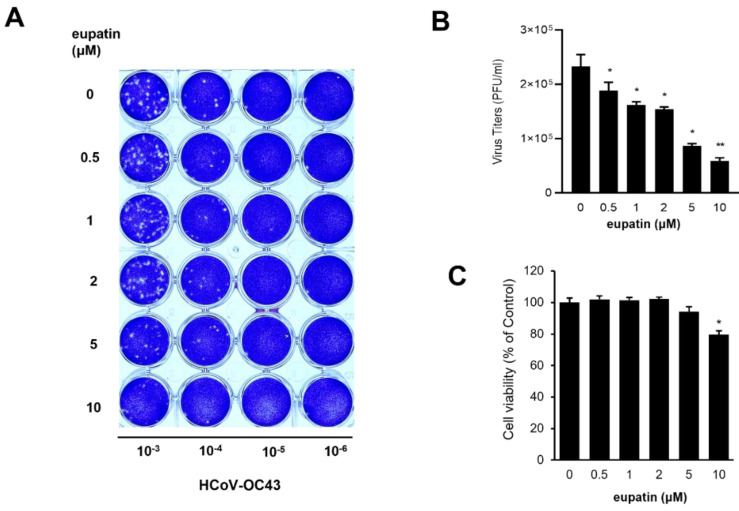
Treatment with eupatin from *I*. *britannica* flowers decreases coronavirus replication. (**A**) RD cells were infected with mock and HCoV-OC43. Plaque assays were performed with agarose overlay and incubated for 3 days. The infected cells were stained with crystal violets to visualize the viral plaques. The dilution factor of HCoV-OC43-conditioned media is shown at the bottom. (**B**) The graph shows mean and standard error. Mock vs. Eupatin treatment (N = 4): * *p* < 0.05, ** *p* < 0.001. (**C**) Eupatin doses of <10 μM did not result in cytotoxicity. Cell viability was evaluated with EZ-cytox. The graph shows mean and standard error. Mock vs. Eupatin treatment (N = 4): * *p* < 0.05.

**Figure 4 ijms-24-09211-f004:**
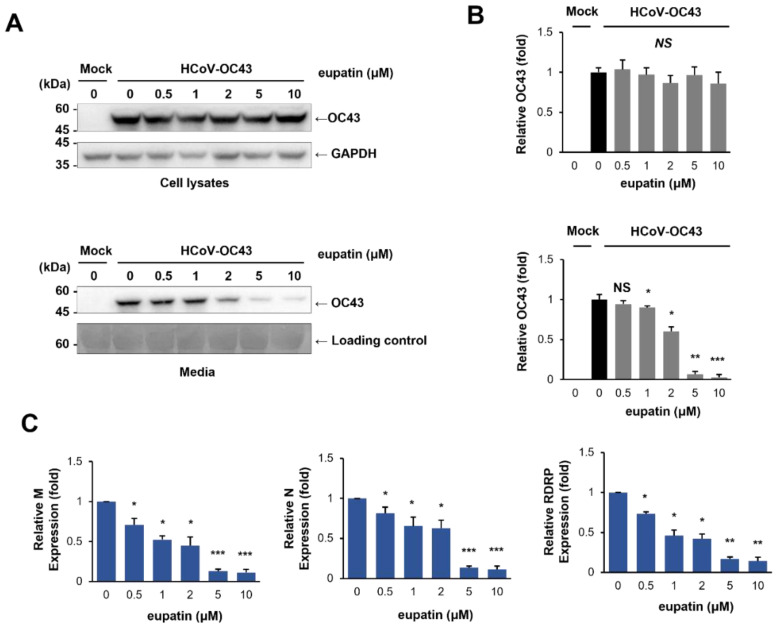
Eupatin inhibits human coronavirus replication in a dose-dependent manner. (**A**) Eupatin treatment decreases the expression of HCoV-OC43 protein in conditioned media. RD cells were incubated with the indicated concentration of eupatin and infected with HCoV-OC43 (10^−3^ dilution of conditioned media). Three days after infection, cell lysates (upper panel) and conditioned media (lower panel) were subjected to western blot with the indicated antibodies. GAPDH western blot was used as a loading control for cell lysates, and Ponceau S stained membrane was used as a loading control for media. (**B**) The expression of OC43 was quantified. The experiment was performed in triplicate. The graph displays the mean and standard error. Mock vs. Eupatin treatment; * *p* < 0.05, ** *p* < 0.005, *** *p* < 0.001, NS—not significant. (**C**) RD cells were infected with HCoV-OC43, and RNA was collected from infected cell media. The RNA levels of the M, N, and RDRP genes were assessed by quantitative RT-PCR. The experiments were performed in triplicate, and the graph shows mean and standard error. Mock vs. Eupatin treatment; * *p* < 0.05, ** *p* < 0.005, *** *p* < 0.001, NS—not significant.

**Table 1 ijms-24-09211-t001:** The inhibitory activity of isolated eupatin from *I. japonica* on 3CLpro.

IC_50_ (μM)	Binding Type (*k*_i_, μM)	Hydrogen Bonds	Autodock Score (kcal/mol)
34.9 ± 1.7	Competitive (30)	Cys145 (2.91), Arg188 (2.79), Thr190 (2.98, 2.68), Gln192 (2.95)	−7.47

## Data Availability

Not applicable.

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
