# Peer review of "Eupatin, a Flavonoid, Inhibits Coronavirus 3CL Protease and Replication"

_ijms, 2023, doi:10.3390/ijms24119211_

Round 1

Reviewer 1 Report (New Reviewer)

In this study, the authors isolate the flavonoid eupatin from I. japonica flowers and tested it for inhibition of the SARS-CoV-2 protease 3CL. They used molecular docking techniques and other simulation techniques to demonstrate a probable docking site on the surface of the protease. Then they tested the effects of eupatin on SARS-CoV-2 replication and found that it significantly decreased the amount of viral proteins (and presumably infectious virions) released from cells. The story fits nicely into the field of chemical biology and the existence of countless biologically active natural products that could be adapted and employed to treat or prevent disease. This paper will be of interest to a broad array of individuals from various scientific disciplines given that it interfaces with herbal medicine, virology, public health, chemical biology, biochemistry, and other fields. 

The paper is extremely well-written, concise, and the results support the conclusions. The techniques are appropriate and the presentation is top-notch. I recommend it for immediate publication.

Author Response

In this study, the authors isolate the flavonoid eupatin from I. japonica flowers and tested it for inhibition of the SARS-CoV-2 protease 3CL. They used molecular docking techniques and other simulation techniques to demonstrate a probable docking site on the surface of the protease. Then they tested the effects of eupatin on SARS-CoV-2 replication and found that it significantly decreased the amount of viral proteins (and presumably infectious virions) released from cells. The story fits nicely into the field of chemical biology and the existence of countless biologically active natural products that could be adapted and employed to treat or prevent disease. This paper will be of interest to a broad array of individuals from various scientific disciplines given that it interfaces with herbal medicine, virology, public health, chemical biology, biochemistry, and other fields. 

The paper is extremely well-written, concise, and the results support the conclusions. The techniques are appropriate and the presentation is top-notch. I recommend it for immediate publication.

-> I appreciate your review and comment.

Reviewer 2 Report (New Reviewer)

The manuscript entitled: Eupatin, a flavonoid, inhibits coronavirus 3CL protease and replication by Park et al., demonstrate the mechanistic details of anti-SARS-CoV-2 activity of Eupatin. Overall the study is exciting and the manuscript is very well-written. I have a few minor concerns. Please find my comments below.  What was the loading control used in Figure 4 A media-treated samples?  A better GAPDH image will be helpful.  Authors should separate RT-PCR and western blot methods with further elaboration of western blotting conditions. The authors should add the catalog numbers and dilutions of all the antibodies used.   Will Eupatin have similar antiviral potential against other strains of the viruses?  

Author Response

The manuscript entitled: Eupatin, a flavonoid, inhibits coronavirus 3CL protease and replication by Park et al., demonstrate the mechanistic details of anti-SARS-CoV-2 activity of Eupatin. Overall the study is exciting and the manuscript is very well-written. I have a few minor concerns.

Please find my comments below. 

What was the loading control used in Figure 4 A media-treated samples?  A better GAPDH image will be helpful. 

-> We used GAPDH western blot as a loading control for cell lysates, however GAPDH cannot be used for media western blot. Therefore, we used the stained membrane as a loading control for media. We added the information in Figure 4.

Authors should separate RT-PCR and western blot methods with further elaboration of western blotting conditions.

-> As the reviewer recommended, we separated the methods for Western blotting in the revised manuscript. We also added the information for western blot.

The authors should add the catalog numbers and dilutions of all the antibodies used.

-> In the revised manuscript, we indicated the catalog number and dilution of OC-43 antibody.

Will Eupatin have similar antiviral potential against other strains of the viruses?

-> Antivirals are normally specific to certain strains of virus, because the antivirals target the specific targets (usually enzymes). Coronavirus contains 3CL-protease, which is conserved in other coronaviruses, therefore eupatin will be effective to coronaviruses. However, eupatin will not show antiviral potential to other strains of viruses, which do not express 3CL-protease.   

Reviewer 3 Report (New Reviewer)

The authors investigated the ability of eupatin to inhibit the replication of the coronavirus, precisely by inhibiting one of the 3 major drug targets of the virus, the 3CL-protease.

Thus, the study is of interest to science because the coronavirus is in continuous modification and adaptation to vaccines and drugs, therefore new and new ways of approach and treatment are needed.

The use of eupatin for this purpose appears from the very beginning to be feasible because it is proven with increased bioavailability on the human intestinal cells (as resulted from studies on Caco-2 cells), at the same time it has anti-inflammatory effects that help to reduce the destructive effects of the coronavirus on the attacked organs.

Overall, the purpose and objectives of the article are well argued, the working methods are clear and efficient, the results are explained and processed statistically. The conclusions are appropriate.

Yet, there is a need for a comparison with the efficiency of other (vegetable) compounds studied through a similar approach, both in terms of the in silico and in vitro approaches.

Also, Figures 1 and 2 are very effective from the point of view of the cumulative information brought, but they must be enlarged for better visibility.

Author Response

The authors investigated the ability of eupatin to inhibit the replication of the coronavirus, precisely by inhibiting one of the 3 major drug targets of the virus, the 3CL-protease.

Thus, the study is of interest to science because the coronavirus is in continuous modification and adaptation to vaccines and drugs, therefore new and new ways of approach and treatment are needed.

The use of eupatin for this purpose appears from the very beginning to be feasible because it is proven with increased bioavailability on the human intestinal cells (as resulted from studies on Caco-2 cells), at the same time it has anti-inflammatory effects that help to reduce the destructive effects of the coronavirus on the attacked organs.

Overall, the purpose and objectives of the article are well argued, the working methods are clear and efficient, the results are explained and processed statistically. The conclusions are appropriate.

Yet, there is a need for a comparison with the efficiency of other (vegetable) compounds studied through a similar approach, both in terms of the in silico and in vitro approaches.

-> Thank you for the comment. To develop coronavirus therapy, many plant-derived compounds were studied, and in silico and in vitro methods were used to show anti-viral effects against coronavirus. Therefore, the extensive review paper will be required to compare the efficacy of plant derived compounds, and it is out of the scope of current manuscript.

Also, Figures 1 and 2 are very effective from the point of view of the cumulative information brought, but they must be enlarged for better visibility.

-> As the reviewer recommended, we enlarged Figure 1 and 2 in the revised manuscript.

This manuscript is a resubmission of an earlier submission. The following is a list of the peer review reports and author responses from that submission.

Round 1

Reviewer 1 Report

Line 53: Scientific names must be in italics

Line 55: Correct and review the entire manuscript (Figure S1A, it is different from Figure 1, which is the correct way to represent it).

Line 86: correct the negative symbol (̶ 7.47 kcal/mol), it is as “_”

Line 215: The temperature symbol must be separated from the unit. Review and, where appropriate, correct the entire text. Equals percent symbol.

Line 303: Check that the references are adapted to the journal, Error (<i>In Vitro</i> A).

In Figure 1, the description of E and F is not described.

I suggest that in section 4.7. Statistical analysis uses statistical programs, there are open codes such as: R, Jasp, Jamovi, or PSPP.

To visualize the molecular coupling (Figure 1F), he does not mention which program he used. Neither the source nor how he obtained the image of figure 1E

I suggest increasing the number of references

Author Response

Line 53: Scientific names must be in italics

Answer: Thank you for the comment, we used the scientific names in italics in the revised manscript.

Line 55: Correct and review the entire manuscript (Figure S1A, it is different from Figure 1, which is the correct way to represent it).

Answer: It was a typo error and we deleted (Figure S1A) in the revised manuscript.

Line 86: correct the negative symbol (̶ 7.47 kcal/mol), it is as “_”

Answer:  Thank you for the comment. We replaced it in the revised manuscript.

Line 215: The temperature symbol must be separated from the unit. Review and, where appropriate, correct the entire text. Equals percent symbol.

Answer: We fixed it in the revised manuscript as the reviewer recommended.

Line 303: Check that the references are adapted to the journal, Error (<i>In Vitro</i> A).

Answer: We checked and fixed it in the revised manuscript.

In Figure 1, the description of E and F is not described.

Answer: In the revised manuscript, we added the description in section 2.3.

I suggest that in section 4.7. Statistical analysis uses statistical programs, there are open codes such as: R, Jasp, Jamovi, or PSPP.

Answer: Thank you for your suggestions. We used simple statistical analysis for the current study, and Microsoft Excel is sufficient and commonly used in this kind of analysis.

To visualize the molecular coupling (Figure 1F), he does not mention which program he used. Neither the source nor how he obtained the image of figure 1E

Answer: Reference #20 in 4.4 section contains the information, and please check the reference #20.

I suggest increasing the number of references

Answer: As the reviewer recommended, we added more than 10 references in the revised manuscript.

Reviewer 2 Report

The manuscript reports a comprehensive study of the inhibition of SARS-CoV-2 3CLpro by eupatin, a flavonoid found in many natural sources. To date, many publications have shown the possibility of identifying antiviral compounds from natural sources, and this manuscript adds another interesting compound. However, the authors have not adequately accounted for other similar compounds and failed to provide an appropriate comparison. Still, the experimental strategy is adequate and the results seem interesting.

Major points:

1. In Figure 1B, is the line the result of a non-linear fitting analysis? If so, what inhibition model or fitting curve did the authors employ?

2. In Figure 1C and 1D, Lineweaver-Burk and Dixon plots are shown. It is well-known that those linearization transformations should only be employed for visualization purposes, and not for data analysis. Data analysis (estimation of IC50 and Ki) should be dome with raw (untransformed) data. Those transformations do not comply with the assumptions in least-squares regression analysis. I addition, and more importantly, those linearization transformations require using the total inhibitor concentration as the x-variable for each data point, whereas the x-variable should be the "free" inhibitor concentration after considering inhibitor depletion due to binding to the enzyme.

3. In Table 1, IC50 and Ki should have units of μM. In the same table, estimated values for IC50, ki, and Autodock score should have errors rounded-off to a single non-zero significant digit, and the estimated value should also be rounded-off according to the error.

4. Figure 3 should not be located before section 2.5. In general, location of Figures is strange: they appear before they are mentioned.

5. As indicated before, many publications have shown the possibility of identifying antiviral compounds from natural sources. And many of these identified compounds are very similar to eupatin: quercetin, myricetin, rutin... The authors should cite some of these references, comparing the structural and functional (i.e., inhibition potency, solubility...) of eupatin with those similar compounds.

6. Which substrate was employed for the inhibition assays? Which is the Km of the substrate? How was the Ki calculated from the IC50?

7. How was 3CLpro obtained?

Author Response

The manuscript reports a comprehensive study of the inhibition of SARS-CoV-2 3CLpro by eupatin, a flavonoid found in many natural sources. To date, many publications have shown the possibility of identifying antiviral compounds from natural sources, and this manuscript adds another interesting compound. However, the authors have not adequately accounted for other similar compounds and failed to provide an appropriate comparison. Still, the experimental strategy is adequate and the results seem interesting.

Major points:

  1. In Figure 1B, is the line the result of a non-linear fitting analysis? If so, what inhibition model or fitting curve did the authors employ?

Answer:  

Equation : y=y+[(ax)/(b+x)], Figure 1B is the result of converting the x-axis to a log value in the equation.

  1. In Figure 1C and 1D, Lineweaver-Burk and Dixon plots are shown. It is well-known that those linearization transformations should only be employed for visualization purposes, and not for data analysis. Data analysis (estimation of IC50 and Ki) should be dome with raw (untransformed) data. Those transformations do not comply with the assumptions in least-squares regression analysis. I addition, and more importantly, those linearization transformations require using the total inhibitor concentration as the x-variable for each data point, whereas the x-variable should be the "free" inhibitor concentration after considering inhibitor depletion due to binding to the enzyme.

Answer: Please refer to the picture below. Confirm the binding method of enzyme-inhibitor+substrate from the Lineweaver-Burk plot (x-axis is “1/substrate concentration”, y-axis is “vo”, initial velocity.)

In the Dixon plot, the intersection point of the linear equation according to each substrate concentration is the ki value (x-axis is “inhibitor concentration”, y-axis is “vo”, initial velocity.).

Lineweaver-burk plots

Dixon plots

  1. In Table 1, IC50 and Ki should have units of μM. In the same table, estimated values for IC50, ki, and Autodock score should have errors rounded-off to a single non-zero significant digit, and the estimated value should also be rounded-off according to the error.

Answer: We revised IC50 and Ki values. Since autodock score is calculated by the autodock program, the S.D value is constant “0”.

  1. Figure 3 should not be located before section 2.5. In general, location of Figures is strange: they appear before they are mentioned.

Answer: As the reviewer recommended, we moved the location of Figure 3 and Figure 4 in the revised manuscript.

  1. As indicated before, many publications have shown the possibility of identifying antiviral compounds from natural sources. And many of these identified compounds are very similar to eupatin: quercetin, myricetin, rutin... The authors should cite some of these references, comparing the structural and functional (i.e., inhibition potency, solubility...) of eupatin with those similar compounds.

Answer: As the reviewer recommended, we added the description with references in discussion (line 193~199).

  1. Which substrate was employed for the inhibition assays? Which is the Km of the substrate? How was the Ki calculated from the IC50?

Answer:  Please refer to the above mentioned in answer 2.

  1. How was 3CLpro obtained?

Answer: We added 3CLpro information in 4.1 section in the revised manuscript.

Round 2

Reviewer 2 Report

The authors have revised the manuscript appropriately. But, still, there is a severe concern that has not been adequately addressed.

As indicated before, in Figure 1C and 1D, Lineweaver-Burk and Dixon plots are shown. It is well-known that those linearization transformations should only be employed for visualization purposes, and not for data analysis. Data analysis (estimation of IC50 and Ki) should be done with raw (untransformed) data applying non-linear least-squares regression data analysis. Those transformations do not comply with the assumptions in least-squares regression analysis. I addition, and more importantly, those linearization transformations require using the total inhibitor concentration as the x-variable for each data point, whereas the x-variable should be the "free" inhibitor concentration after considering inhibitor depletion due to binding to the enzyme.

The authors have just explained the generalities about Lineweaver and Dixon plots, but they have not redone the data analysis. I know the Lineweaver and Dixon plots are still employed for data analysis, but they represent outdated and incorrect procedures.

Author Response

Comments from the editors and reviewers:
-Reviewer 2

  1. In Figure 1C and 1D, Lineweaver-Burk and Dixon plots are shown. It is well-known that those linearization transformations should only be employed for visualization purposes, and not for data analysis. Data analysis (estimation of IC50 and Ki) should be dome with raw (untransformed) data.

Answer: Thank you for the comment. The result of current study, the ki value was derived in the same way as in the following references.

<Reference>

[1] Int. J. Mol. Sci. 2023, 24(3) 2454.

[2] Inter. J. Biol. Macromol. 2020, 165 1822-1831.

[3] Int. J. Mol. Sci. 2013, 14 9873-9882.

Those transformations do not comply with the assumptions in least-squares regression analysis. I addition, and more importantly, those linearization transformations require using the total inhibitor concentration as the x-variable for each data point, whereas the x-variable should be the "free" inhibitor concentration after considering inhibitor depletion due to binding to the enzyme.

Answer: In this study, the x-axis of the Dixon plot used the inhibitor concentration. Moreover, “Free” inhibitor concentration cannot be determined from enzyme kinetics in our study and references.

We believe that the reviewer's comment is a comment on another experiment (binding affinity assay).
